# Muller’s Ratchet and Ribosome Degeneration in the Obligate Intracellular Parasites *Microsporidia*

**DOI:** 10.3390/ijms19124125

**Published:** 2018-12-19

**Authors:** Sergey V. Melnikov, Kasidet Manakongtreecheep, Keith D. Rivera, Arthur Makarenko, Darryl J. Pappin, Dieter Söll

**Affiliations:** 1Department of Molecular Biophysics and Biochemistry, Yale University, New Haven, CT 06511, USA; serguey.v.melnikov@gmail.com (S.V.M); kasi.mana37@gmail.com (K.M.); 2Cold Spring Harbor Laboratory, Cold Spring Harbor, NY 11724, USA; krivera@cshl.edu (K.D.R); amakaren@cshl.edu (A.M.); pappin@cshl.edu (D.J.P.); 3Department of Molecular Biophysics and Biochemistry, Department of Chemistry, Yale University, New Haven, CT 06511, USA

**Keywords:** Muller’s ratchet, genome decay, ribosome, rRNA expansions, rudimentary proteins

## Abstract

*Microsporidia* are fungi-like parasites that have the smallest known eukaryotic genome, and for that reason they are used as a model to study the phenomenon of genome decay in parasitic forms of life. Similar to other intracellular parasites that reproduce asexually in an environment with alleviated natural selection, *Microsporidia* experience continuous genome decay that is driven by Muller’s ratchet—an evolutionary process of irreversible accumulation of deleterious mutations that lead to gene loss and the miniaturization of cellular components. Particularly, *Microsporidia* have remarkably small ribosomes in which the rRNA is reduced to the minimal enzymatic core. In this study, we analyzed microsporidian ribosomes to study an apparent impact of Muller’s ratchet on structure of RNA and protein molecules in parasitic forms of life. Through mass spectrometry of microsporidian proteome and analysis of microsporidian genomes, we found that massive rRNA reduction in microsporidian ribosomes appears to annihilate the binding sites for ribosomal proteins eL8, eL27, and eS31, suggesting that these proteins are no longer bound to the ribosome in microsporidian species. We then provided an evidence that protein eS31 is retained in *Microsporidia* due to its non-ribosomal function in ubiquitin biogenesis. Our study illustrates that, while *Microsporidia* carry the same set of ribosomal proteins as non-parasitic eukaryotes, some ribosomal proteins are no longer participating in protein synthesis in *Microsporidia* and they are preserved from genome decay by having extra-ribosomal functions. More generally, our study shows that many components of parasitic cells, which are identified by automated annotation of pathogenic genomes, may lack part of their biological functions due to continuous genome decay.

## 1. Introduction

It is well documented that the parasitic lifestyle causes continuous genome decay, manifested in progressive gene loss and accumulation of deleterious mutations in essential genes [1,2,3]. Yet we are only beginning to understand how genome decay impacts structure of individual proteins and nucleic acids in parasitic organisms.

Among numerous forms of life that inhabit our planet, there is a group of single-cell pathogens that can grow and reproduce only inside other living cells. These organisms evolve under conditions that are radically different from those of free-living species: they live in small population with little competition for nutrients and shelter, which weakens natural selection; they undergo repeated population bottlenecks upon transition from one host cell to another, which favors genetic drifts; and, they proliferate asexually, which prevents the elimination of toxic mutations through recombination of genetic material [3]. Consequently, obligate intracellular parasites evolve under conditions that favor gradual and irreversible accumulation of deleterious mutations. In the long-term, this irreversible process, known as Muller’s ratchet, causes genome decay that is reflected in the massive loss of non-essential genes and an increasing number of deleterious mutations in essential genes in parasite genomes [3,4].

The genome decay that is caused by Muller’s ratchet can be observed at every level of cellular organization, from organelles to individual proteins and nucleic acids. For instance, if we take genome size in bacterial species, more than over 4000 protein-coding genes are typically found in species with free-living lifestyle; approximately ~2000–3000 protein-coding genes in species with hybrid lifestyle with elements of free-living and parasitism, less than ~1500 of protein-coding genes are found in most parasitic species with strictly intracellular lifestyle. In some extreme cases of long-term intracellular parasitism, the total number of genes can be as low as 500 protein-coding genes [5]. The remaining essential genes in parasitic organisms undergo truncations and accumulate deleterious mutations that compromise protein and RNA folding, reduce enzymatic activity, and possibly reduce the specificity of molecular interactions inside parasitic cells [6,7,8].

An increasing interest to the phenomenon of genome decay stems from a growing evidence that this erosive evolutionary process might be used as an “Achilles’ heel” to create new therapies against parasitic infections [7,9,10,11]. For instance, massive loss of genes that are involved in nutrient biosynthesis, nutrient sensing, stress response, and quality control can make parasitic cells vulnerable to stressors and factors that require activity of quality control systems [7,9]. Or accumulation of deleterious mutations in essential proteins and nucleic acids may make their structures more sensitive to heat and more dependent on the activity of molecular chaperones, suggesting the hypersensitivity of parasites to heat shock and chaperone inhibitors [10,11,12,13,14]. Importantly, the genome decay in parasitic forms of life is observed in intracellular pathogens from all kingdoms of life, regardless of their evolutionary origin [2,5]. Hence, if therapeutic targeting of decayed proteins and nucleic acids can be found for one group of parasites, it can be potentially applied to a much broader range of parasitic species.

One clinically relevant model to study extreme genome decay can be observed in the fungi-related pathogens *Microsporidia*. These organisms have the smallest known eukaryotic genome, with many microsporidian species comprising as little as ~1800 genes, which is ~3 times smaller than genomes of non-parasitic fungi, such as the yeast *Saccharomyces cerevisiae* [15,16]. *Microsporidia’s* miniature bodies are made of a single cell whose size is comparable to that of *Escherichia coli*. All known *Microsporidia* species are parasites; several species, including *Enterocytozoon bieneusi*, *Encephalitozoon cuniculi*, and *Trachipleistophora hominis*, are found in ~10–40% HIV-positive humans, where they cause chronic infections of the gastrointestinal tract, eye, and kidney [17]. Due to lack of effective treatment against these pathogens, they are included in the list of priority emerging pathogens by the National Institute of Allergy and Infectious Diseases [18].

As is the case in other intracellular parasites, regressive changes in *Microsporidia* can be observed at every scale of their organization, from their overall cell structure to the structure of individual macromolecules [15,16]. The most impressive regressions are observed in the protein synthesis machinery, particularly in the ribosome. When compared to ribosomes from other eukaryotes, whose molecular weight varies between ~3.3 MDa and ~4.5 MDa, microsporidian ribosomes are expected to weigh only ~2.2 MDa, which is even smaller than ~2.5 MDa ribosomes from bacteria [19,20,21]. This remarkable reduction is achieved primarily by the massive degeneration of microsporidian rRNA, particularly the degeneration of rRNA expansions (rRNA segments that are missing in bacteria, but present in most eukaryotes, where many of them are thought to be involved in regulation of protein synthesis). For instance, in most microsporidian species, the 18S rRNA length varies around ~1250–1350 bases, as compared to ~1800–1850 bases in non-parasitic eukaryotes [20,21,22,23]. This is especially impressive given the fact that even in bacterial ribosomes the 16S rRNA comprises ~1600 bases. It is therefore assumed that microsporidian ribosomes represent the minimal machinery of protein synthesis stripped down from regulatory and quality control-related components and debased to the minimal enzymatic core [15,20].

To better understand how parasitic lifestyle affects structure of proteins and nucleic acids, we explored in this study the fate of ribosomal proteins in microsporidian parasites. Particularly, we asked how the massive loss of rRNA expansions could have affected the ribosomal proteins that are bound to rRNA expansions in ribosomes from non-parasitic eukaryotes. By using comparative analysis of microsporidian genome sequences and the analysis of structures of fungal ribosomes, we showed that the massive reduction of microsporidian rRNA appears to eliminate the interface between ribosomal RNA and several ribosomal proteins, including eS31, eL8, and eL27. These ribosomal proteins appear to lose their attachment to the ribosome and turn into free-standing proteins in microsporidian parasites. We further used mass spectrometry analysis of proteins from a microsporidian *E. cuniculi* to show that eS31, eL8, and eL27 proteins are still abundantly expressed in the microsporidian proteome. For protein eS31, we show that its preservation in the microsporidian proteome is likely related to the extraribosomal function of this protein in ubiquitin biogenesis. In sum, our study illustrates that, despite the seeming conservation in parasites with highly reduced genomes, some components of the protein synthesis machinery could have lost their original role in protein synthesis and they are retained in parasites by having non-protein synthesis related activities.

## 2. Materials and Methods

### 2.1. Analysis of rRNA-Protein Contacts Evolution in the Ribosome Structure

To illustrate rRNA degeneration in microsporidian parasites, we compared the secondary structure diagrams of rRNA from the microsporidian *Encephalitozoon cuniculi* and the yeast *Saccharomyces cerevisiae.* For both species, the diagrams of 18S, 5.8S, and 25S rRNA were retrieved from the Comparative RNA Website (http://www.rna.icmb.utexas.edu/) (access on 12 November 2018) and from the previous studies of *E. cuniculi* rRNA degeneration [20,21]. Subsequently, the crystal structure of the 80S *S. cerevisiae* ribosome (pdb id 5dgv) was used to map rRNA segments that are present in *S. cevevisiae* but degenerated in *E. cuniculi*.

The rRNA-protein interface was calculated by using the yeast 80S ribosome structure (pdb id 5dgv) and an online tool to measure protein interfaces in macromolecular assemblies jsPISA from the CPP4 package (http://www.ccp4.ac.uk/pisa/) (access on 12 November 2018) [24]. The conservation of protein sequences was estimated by using multiple sequence alignments with ClustalO [25]. Microsporidian species that were used in the analysis of eS31 sequence conservation are *Nosema ceranae*, *Edhazardia aedis*, *Spraguea lophii*, *Trachipleistophora hominis*, *Vavraia culicis*, *Vittaforma corneae*, *Nematocida ausubeli*, *Nematocida parisii*, *Nematocida displodere*, *Enterocytozoon hepatopenaei*, *Encephalitozoon cuniculi*, *Mitosporidium daphnia*, and *Pseudoloma neurophilia* (note that in some species ubiquitin-eS31 is annotated as ubiquitin-S27a or ubiquitin-carboxy-terminal extension protein). Yeast sequences that were used in the analysis of eS31 sequence conservation are *Kluyveromyces lactis*, *Schizosaccharomyces pombe*, *Scheffersomyces stipites*, *Exophiala dermatitidis*, *Coniosporium apollinis*, *Candida auris*, *Pseudozyma antarctica*, *Ogataea parapolymorpha*, *Wickerhamomyces ciferrii*, *Hanseniaspora uvarum*, *Pichia kudriavzevii*, *Tetrapisispora phaffii*, and *Torulaspora delbrueckii*.

### 2.2. Preparation of Protein Extracts from Microsporidian Spores

Microsporidian spore production and protein extraction was adapted from [26]. The *E. cuniculi* isolate GB-M1 used for genome sequencing was cultured in Madin-Darby canine kidney or human foreskin fibroblast cells, as previously described [27]. After collection of culture supernatants, parasites were sedimented (5000 g, 5 min) and heated at 65 °C in a 1% SDS solution. A spore-rich fraction was recovered after several washes in water to remove host cell debris. Parasites were disrupted in a lysis buffer (10^9^–10^10^ cells/mL) containing 100 mM DTT, 4% CHAPS, and 0.2% SDS, by repeated cycles of freezing-thawing and sonication (Deltasonic 1320, 300 W, 28 kHz). Proteins of broken cells were then extracted with a solution containing 7 M urea, 2 M thiourea, 100 mM DTT, 4% CHAPS and 0.2% SDS for 5 h at room temperature. After centrifugation (16,100 g, 5 min), the supernatant was collected. The protein sample was characterized by SDS-PAGE on a 12% polyacrylamide gel and stored at ‒20 °C.

Tryptic digestion of microsporidian protein extracts. Tris (2-carboxyethyl)phosphine (TCEP) was added to final concentration of 5mM. Samples were then heated to 55 °C for 20 min, allowed to cool to room temperature, and methyl methanethiosulfonate (MMTS) was added to a final concentration of 10 mM. Samples were incubated at room temperature for 20 min to complete the blocking of free sulfhydryl groups. Lysates were acidified with phosphoric acid to a final concentration of 1.2% and added to an S-Trap^TM^ containing 6x lysate volume of s-trapping buffer (90% methanol, 100 mM TEAB). S-Trap^TM^ was spun down at 4000 g for 30 s to remove buffer, washed with 0.2 mL of S-trapping buffer, and spun again to remove all buffer. 2 µg of sequencing grade trypsin (Promega) in 125 µL of 50 mM TEAB was then added to the S-Trap^TM^ and they were digested overnight at 37 °C. After digestion, the peptides were eluted from the column with subsequent applications of 50 mM TEAB, 0.2% formic acid in water, and 0.2% formic acid in 50% acetonitrile. Peptides were dried in vacuo. Peptides were then reconstituted in 50 µL of 0.5 M TEAB/70% isopropanol and labeled with 8-plex iTRAQ reagent for 2 h at room temperature, essentially according to [28]. Labeled samples were then acidified to pH 4 using formic acid, combined and concentrated in vacuum until ~10 µL remained [29,30].

### 2.3. Mass Spectrometry

An Orbitrap Fusion Lumos mass spectrometer (Thermo Fisher Scientific, Waltham, MA, USA) equipped with a nano-ion spray source coupled to an EASY-nLC 1200 system (Thermo Fisher Scientific, Waltham, MA, USA) was used. The LC system was configured with a self-pack PicoFrit™ 75-μm analytical column with an 8-μm emitter (New Objective, Woburn, MA, USA) packed to 25 cm with ReproSil-Pur C18-AQ, 1.9 µM material (Dr. Maisch GmbH). Mobile phase A consisted of 2% acetonitrile; 0.1% formic acid and mobile phase B consisted of 90% acetonitrile; 0.1% formic acid. Peptides were then separated using the following steps: at a flow rate of 200 nL/min: 2% B to 6% B over 1 min, 6% B to 30% B over 84 min, 30% B to 60% B over 9 min, 60% B to 90% B over 1 min, held at 90% B for 5 min, 90% B to 50% B over 1 min, and then flow rate was increased to 500 nL/min as 50% B was held for 9 min. Eluted peptides were directly electrosprayed into the Fusion Lumos mass spectrometer with the application of a distal 2.3 kV spray voltage and a capillary temperature of 300 °C. Full-scan mass spectrum (Res = 60,000; 400–1600 m/z) was followed by MS/MS using the “Top N” method for selection. High-energy collisional dissociation (HCD) was used with the normalized collision energy set to 35 for fragmentation, the isolation width set to 1.2 and a duration of 10 s was set for the dynamic exclusion with an exclusion mass width of 10 ppm. We used monoisotopic precursor selection for charge states 2+ and greater, and all data were acquired in profile mode.

### 2.4. Database Searching to Detect Protein Identity in Microsporidian Bulk Protein Extracts

The total of 44,547 peptides were used in the analysis. Peaklist files were generated by Mascot Distiller (Matrix Science) [31]. Protein identification and quantification was carried using Mascot 2.450 against the Uniprot_*Microsporidia* database (2041 sequences; 694,360 residues). Methylthiolation of cysteine and N-terminal and lysine iTRAQ modifications were set as fixed modifications and methionine oxidation was set as variable. Trypsin was used as cleavage enzyme with two missed cleavages being allowed. Mass tolerance was set at 5 ppm for intact peptide mass and 0.07 Da for fragment ions. Search results were rescored to give a final 1% FDR using a randomized version of the same Uniprot *Microsporidia* database.

## 3. Results

### 3.1. Massive Loss of rRNA Expansion Segments Degenerates Binding Sites for Ribosomal Proteins in Microsporidian Ribosomes

We first asked how the loss of rRNA expansions could have affected rRNA-protein contacts in microsporidian ribosomes. To anwer this questions, we analyzed rRNA-protein contacts in the 80 S ribosome structure from yeast *S. cerevisiae*, which are ones of the closest organisms to *Microsporidia* among species with known ribosomes structure .

As shown previously [20,21], microsporidian rRNA lacks all major rRNA expansion segments (Figure 1). Mapping of rRNA deletions on the three-dimensional (3D) ribosome structure showed that microsporidian ribosomes additionally lack a few conserved rRNA segments, including helices h15, h16, h17, and h33 in the 18S rRNA, which are otherwise conserved in species from bacteria to eukaryotes (Figure 1A).

Deletions in rRNA in microsporidian ribosomes occur on the ribosome suface, where rRNA expansions extensively interact with ribosomal proteins (Figure 1A,B). Our analysis of rRNA-protein interactions in the 80S yeast ribosome showed that the loss of rRNA expansions should eliminate at least 1/3 of the protein-ribosome interface for eight ribosomal proteins (Figure 1B). In one extreme case, the truncation of rRNA expansions ES27L and ES31L in the 25S rRNA appears to eliminate more than 85% interface between proteins eL27 and other ribosomal components. Similarly, the degeneration of helix h33 in the 18S rRNA appears to eliminate more than 80% interface between protein eS31 and other ribosomal components. More than 50% of the interface between protein eS7 and other ribosomal components appears to get lost due to degeneration of ES3S and ES6S rRNA expansions in microsporidian 18S rRNA. Given this massive loss of protein-rRNA interactions, it is plausible that some ribosomal proteins, particularly eS31 and eL27, represent stand-alone rather than ribosome-bound proteins in microsporidian species.

### 3.2. Proteins that Appear to Lose Association with the Ribosome also Lose Their Sequence Conservation

We next asked if the loss of rRNA expansion in *Microsporidia* is accompanied with loss of corresponding ribosomal proteins that are bound to rRNA expansions in ribosomes from non-parasitic eukaryotes. Through analysis of microsporidian genome sequences, we found that even proteins eS7, eS31, and eL27 are still retained in microsporidian species, although in some species they have undergone significant truncations (Figure 2A–D). This is especially prominent in protein eS31, where the sequence is truncated by ~70% in some microsporidian species, including *Edhazardia aedis* and *Vittaforma corneae* (Figure 2C). However, despite massive truncations in ribosomal proteins in some microsporidian species, even ribosomal proteins that have lost the corresponding rRNA expansion segments are still present in microsporidian species.

We next analyzed sequence conservation of ribosomal proteins eS7, eS31, and eL27. We anticipated that, in Microsporidia, these proteins should be highly variable because they are no longer involved in interactions with rRNA. Indeed, we found that these apparent free-standing ribosomal proteins have poorly conserved sequences in microsporidian species (Figure 2C,D). One especially notable example is protein eS31 (Figure 2C,D). Apart from its role in protein synthesis, eS31 has an additional function in ubiquitin biogenesis: in eukaryotes, eS31 is produced as a fusion with ubiquitin, and ubiquitin is being cleaved off from eS31 during ribosome biogenesis. We found that, in Microsporidia, the ubiquitin sequence remains more than 90% invariant in microsporidian species, whereas the eS31 sequence carries only six invariant residues, illustrating the extremely poor conservation of eS31 in microsporidian species (Figure 2B). Notably, four of these six invariant residues represent cysteines that are critically required for eS31 folding (Figure 2B,C). Particularly, this example of eS31 conservation in microsporidian species indicates that some ribosomal proteins could have lost their original role in protein syntheis and are preserved in microsporidia for their extra-ribosomal functions.

### 3.3. Apparent Rudimentary Ribosomal Proteins are Abundantly Expressed in Microsporidian Cells

We finally asked whether the genes coding for degenerated ribosomal proteins represent functional genes or pseudogenes. To address this, we used mass spectrometry and analyzed protein exctracts from microsporidian *E. cuniculi* spores (Appendix A). We found that all of the apparent free-standing proteins, including eS31, eL27, and eS7, were observed among the most frequently detected proteins in the microsporidian proteome. For instance, protein eS31 (annotated as “Similarity to monoubiquitin/carboxy-extension protein fusion” in the *Enciphalitozoon cuniculi* genome due to poor eS31 sequence conservation) was found among the top 200 most frequently detected proteins (Appendix A). This analysis revealed that, despite the apparent loss of ribosomal association, the ribosomal proteins are still abundantly expressed in microsporidian cells.

## 4. Discussion

In the present study, we explored an apparent effect of Muller’s ratchet on the structure of microsporidian ribosomes. Particularly, we analyzed ribosomal proteins that stabilize and coordinate activity of rRNA expansions in eukaryotic ribosomes. We found that the massive loss of rRNA expansion in microsporidian ribosomes appears to annihilate the binding sites for a few ribosomal proteins (eS7, eS31, and eL27), suggesting that these proteins are no longer part of the protein synthesis machinery, but rather free-standing proteins that are retained in microsporidian species for extra-ribosomal functions, or simply awaiting their total extinction forced by genome decay.

Our study illustrates that automated analysis of parasitic genomes may significantly underestimate the extent of genome decay in parasitic species. We show here that some components of a parasitic cell that are “defined” by automated annotations as functional constituents of the protein synthesis machinery may in fact represent rudimentary proteins that are retained in parasites for their non-ribosomal function, as is the case of microsporidian protein eS31, which is involved in ubiquitin biogenesis. Thus, our study illustrates the presence of evolutionary intermediates between the genes and pseudogenes in parasitic genomes. These genes are not yet pseudogenes, because they are actively expressed into proteins; yet, the proteins that are produced from these genes are not fully active functionally as they lack at least one of their original biological activities in non-parasitic species.

Regarding the ribosome function, it is pertinent to mention that the case of *Microsporidia* shows that the eukaryotic ribosome can function in vivo in the complete absence of rRNA expansion segments. Furthermore, our study illustrates that microsporidian ribosomes not only lack rRNA expansion segments, but a few rRNA segments that are conserved between eukaryotes and bacteria, particularly helix h16 in the 18S rRNA. During protein synthesis, helix h16, among other factors, is thought to promote ribosomes self-organization into polysomes, in which each individual ribosome is properly spaced and oriented to allow for access of regulatory proteins and factors of co-translational protein processing and processing. In eukaryotes, helix h16 also participates in the initiation of protein synthesis by helping the ribosome to find the initiation codon in the messenger RNA [32,33,34]. It is tempting to imagine that the lack of this helix in microsporidian ribosomes leads to a less organized structure of polysomes and more primitive forms of protein synthesis initiation, or even occasional codon recognition errors during the start-codon search in messenger RNA.

Finally, our study illustrates a conceptual similarity of how evolution works at two radically different scales: at the scale of proteins and nucleic acids and at the scale of organs and multicellular organisms (Figure 3A,B). At the scale of organs and organisms, rudimentary features have been well studied in many animal and plant models. For instance, in a classical study of eye evolution in blind mole rats *Spalax ehrebegi*, the regression of eye function was shown to eliminate many anatomical features required for spatial vision (Figure 3B) [35,36]. Particularly, the eye in mole rats gets buried under the skin, and its lens is degenerated to a non-functional rudiment (Figure 3B). Yet, mole rate retain a non-visual eye function: at the bottom of the eye ball mole rates retain a layer of photosensitive cells of retina that allows these blind animals to perceive light and regulate such vital aspects of animal activity as hibernation, thermoregulation, and reproduction (Figure 3B). Our example of ribosomal protein eS31 in microsporidian parasites demonstrates essentially the same regressive evolutionary process as the one that was observed in mole rats’ eye, may be observed at the scale of individual proteins: although eS31 is found in microsporidian genomes, it appears to no longer function as a ribosomal protein, although it may retain non-ribosomal activities (Figure 3A). This finding illustrates that the presence of a protein in a parasite genome does not necessarily means that this protein is fully active. Also, as eS31 case shows, changes in protein structure that are observed in a particular group of parasites may represent not biological adaptations, but a product of genome decay. 

## Figures and Tables

**Figure 1 ijms-19-04125-f001:**
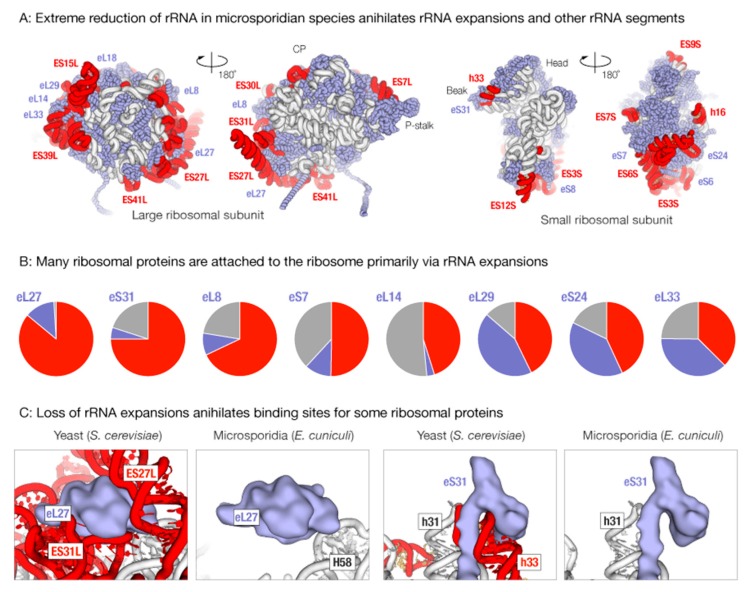
Extreme reduction of microsporidian rRNA appears to annihilate many rRNA-protein contacts, transforming some ribosomal proteins into ribosome-bound or free-standing rudiments with no apparent biological function. (**A**) The panel illustrates massive loss of rRNA expansion segments in microsporidian ribosomes. It shows the ribosome structure from yeast *Saccharomyces cerevisiae*—a non-parasitic fungi that is one of the closest non-parasitic relatives of *Microsporidia* for which there is a known ribosome structure – in which the rRNA segments that are missing in microsporidian species are highlighted in red. The rRNA segments that are conserved between the yeasts and *Microsporidia* are shown in grey, and ribosomal proteins are shown in blue. (**B**) The diagrams summarize the rRNA-protein and protein-protein interface for eight ribosomal proteins that form interact primarily with rRNA expansion segments in the 80S yeast ribosome. (**C**) The panels zoom on the structure of the 80S ribosomes to illustrate that protein eS31 is anchored to the ribosome via its contact with helix h16 in the 16S rRNA, and protein eL27 is sandwiched between the rRNA expansion segments ES27L and ES31L in the 25S rRNA. Because h16, ES27L, and ES31L are fully degenerated in microsporidian rRNA, microsporidian proteins eL27 and eS31 are predicted to no longer be components of protein synthesis machinery and rather be free-standing proteins in *Microsporidia*.

**Figure 2 ijms-19-04125-f002:**
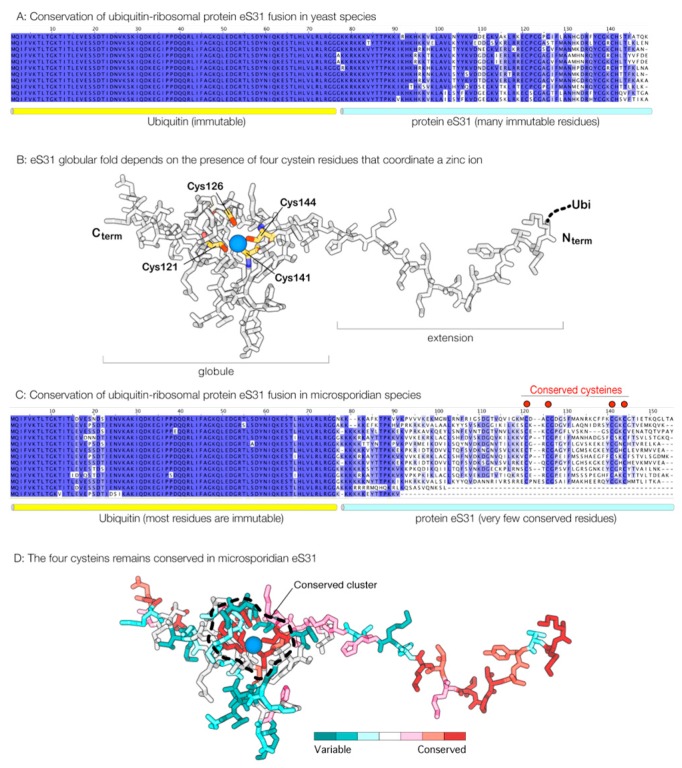
In parasites *Microsporidia*, ribosomal protein eS31 turns into a poorly conserved peptide with only a few residues remaining conserved due to their critical role in protein folding. (**A**) In most eukaryotes, protein eS31 is produced as a fusion with ubiquitin. During ribosome biogenesis, ubiquitin-eS31 fusion is being cleaved into ubiquitin and eS31. The panel illustrates sequence conservation of ubiquitin-eS31 in yeast species, where ubiquitin sequence remains immutable, and eS31 sequence has more than 80% identity between yeast species. (**B**) Crystal structure of eS31 protein shows that the folding of eS31 critically depends on the presence of a zinc-finger motif. The zinc-finger motif is required to stabilize the folding of the eS31 globular domain because the hydrophobic interactions in the globule of this small protein are too weak to keep its polypeptide chain folded. (**C**) In *Microsporidia*, the ubiquitin sequence remains highly conserved, with only ~10% of sequence showing moderate variations, whereas the eS31 sequence becomes into highly variable, with only six residues being conserved between microsporidian species. Four of these residues (all cysteines) coordinate a zinc ion in the middle of eS31 globular domain to form a zinc-finger motif that is used to anchor eS31 in the ribosome structure. (**D**) The crystal structure of protein eS31 is colored by conservation of amino acid residues across microsporidian species (calculated by using the ConSurf server at http://consurf.tau.ac.il/2016/) (access on 12 November 2018). The cluster of conserved residues around zin ion is outlined to show that the most conserved residues are clustering around the zinc ion. High conservation of the four cysteine residues in eS31 in *Microsporidia* suggests that there is selective pressure to keep eS31 properly folded, which is possibly to avoid the formation of protein aggregates that would intoxicate microsporidian cells or would compromise ubiquitin biosynthesis. Overall, the figure illustrates that that, despite an apparent loss by eS31 its original role in protein synthesis, this protein is still retained in the microsporidian proteome for its extraribosomal role in ubiquitin biogenesis.

**Figure 3 ijms-19-04125-f003:**
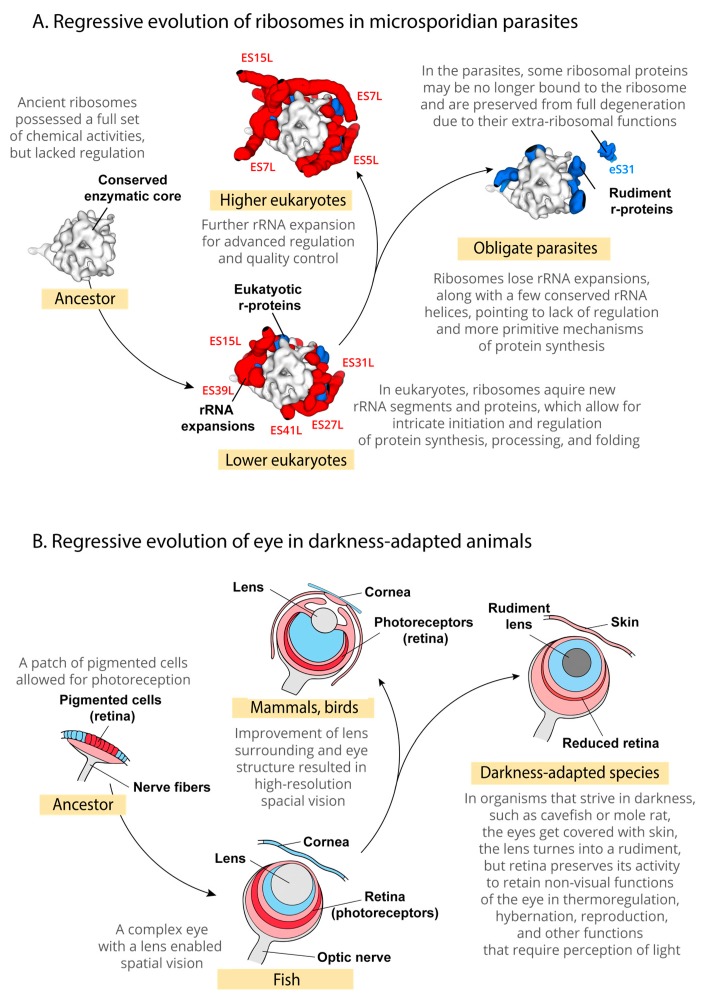
Degeneration of microsporidian ribosomes illustrates a remarkable similarity of regressive evolution at the scale of individual proteins and nucleic acids, and at the scale of organs of animal body. (**A**) Our study suggests that upon transition to parasitic lifestyle, microsporidian ribosomes not only have lost their rRNA expansions, but also transformed several ribosomal proteins from active players of ribosome assembly into dysfunctional rudiments. For instance, in non-parasitic eukaryotes, many ribosomal proteins are buried in the ribosome interior, where they stabilized interactions between the ancient core of the ribosome and rRNA expansions. In *Microsporidia*, these proteins (i.e., eS6, eS8, eS24, eL8, eL14, eL18, eL39) are expected to be exposed on the ribosomal surface where they have no apparent biological function. Other proteins, such as eS31, eL27, and possibly eS7, appear to become free standing proteins that are no longer associated with the ribosome in *Microsporidia*. However, eS31, eL27, and eS7 proteins are not eliminated from microsporidian proteomes, suggesting that these proteins may be retained in *Microsporidia* for their extra-ribosomal functions. (**B**) A schematic path of eye evolution shows that similar regressive changes occur during animal adaptation to darkness, as observed in cavefish and mole rats. Importantly, although the eye of darkness-adapted animals is blind and is filled with many rudiments (e.g., the non-functional degenerated lens), it retains the light-sensitive retina that fulfils the non-visual functions of light perception to control animal hibernation, mating, and thermoregulation [35,36].

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
