# Peer review of "Muller’s Ratchet and Ribosome Degeneration in the Obligate Intracellular Parasites Microsporidia"

_ijms, 2018, doi:10.3390/ijms19124125_

Reviewer 1 Report

The manuscript by Melnikov et al describes how proteins in organsisms under high pressure for genome miniaturization (the effect of which is termed Muller’s ratchet) lose protein functionalities, but not proteins. They demonstrate this using several ribosomal proteins of Microsporidia, which supposedly no longer interact with the ribosome due to rRNA truncation and deletion of protein domains.

The claims put forward in this manuscript are very solidly argued, and backed up with good experimental data. The conclusions are equally sound.

I have two major issues with the manuscript:

First, the manners in which the data is presented, seems off; several results are presented in the Methods section, the Figures are not consecutively labeled or introduced; this goes along with jumps in the argumentation.

Secondly, the discussion of eye protein evolution in darkness-adapted animals has no meaningful intellectual link to the rest of the manuscript. To use it as an additional example provides no additional value to the reader and can simply be left out.

I do think that the overall quality of the manuscript definitely warrants publication, if the concerns raised above are addressed during revision.

Minor comments:

-        Several sentences are poorly or wrongly phrased (i.e. line 268, 309-311)

-        Typos: especially (l256), ubiquitin (l259)

-        The methods section does not define the iTRAQ reagent. S-Trap columns should be combined with a superscript TM

-        The function of the zinc finger in eS31 (lines 138, 145) is poorly explained.

Author Response

First of all, we would like to thank the reviewer for their thorough reading of our manuscript, for the thoughtful comments, and for drawing our attention to multiple errors and shortcomings in our work. We have addressed all the comments and provide below a point-by-point reply.

Reviewer #1

The manuscript by Melnikov et al describes how proteins in organisms under high pressure for genome miniaturization (the effect of which is termed Muller’s ratchet) lose protein functionalities, but not proteins. They demonstrate this using several ribosomal proteins of Microsporidia, which supposedly no longer interact with the ribosome due to rRNA truncation and deletion of protein domains.

The claims put forward in this manuscript are very solidly argued and backed up with good experimental data. The conclusions are equally sound.

I have two major issues with the manuscript:

First, the manners in which the data is presented, seems off; several results are presented in the Methods section, the Figures are not consecutively labeled or introduced; this goes along with jumps in the argumentation.

We have corrected this issue by moving the Figure 2 from the Methods section and by rewriting numerous sentences in the manuscript. All the changes were tracked and are highlighted in the new version of the manuscript.

Secondly, the discussion of eye protein evolution in darkness-adapted animals has no meaningful intellectual link to the rest of the manuscript. To use it as an additional example provides no additional value to the reader and can simply be left out.

We completely agree with the reviewer that our original manuscript didn’t manage to provide a satisfactory explanation of why evolution of proteins in parasites is conceptually similar to organ evolution in animals. In the present manuscript, we have totally rewritten the corresponding paragraph of the text (the last paragraph in the Discussion) and hope that the reviewer will consider our changes satisfactory and sufficient to keep the eye evolution example in the text.

I do think that the overall quality of the manuscript definitely warrants publication, if the concerns raised above are addressed during revision.

Minor comments:

-        Several sentences are poorly or wrongly phrased (i.e. line 268, 309-311)

            The sentences were rewritten.

-        Typos: especially (l256), ubiquitin (l259)

            These and other typos were corrected.

-        The methods section does not define the iTRAQ reagent. S-Trap columns should be combined with a superscript TM.

            The reagent was defined and the TM label was superscripted.

-        The function of the zinc finger in eS31 (lines 138, 145) is poorly explained.

We have added a brief explanation to these lines:

“a zinc-finger motif that is used to anchor eS31 in the ribosome structure”.

Reviewer 2 Report

In this manuscript Melnikov et. al., present data to show that Microsporadia carries same set of ribosomal proteins as other non-parasitic eukaryotes. They further show that, some of the ribosomal proteins no longer associate with ribosome but still they are preserved from genome decay due to their extra-ribosomal functions. They use Genome sequence analysis, non-parasitic ribosome structure analysis, and mass spectrometry to support their conclusions. 

    This is an interesting article and it shows how a minimal Ribosomal unit is fully functional for Microsporadia. Their study shows that, despite reduced genomes in the parasites, some ribosomal proteins which have lose their roles in protein synthesis are still conserved and functional due to their non-ribosomal roles. 

This manuscript may be accepted in its present form. Few minor comments as follows;

Figure 2 should be removed from the Materials and methods section and should be put after Figure 1. 

Figure labeling legends particularly in Figure 2 & 3 are way too small. 

Author Response

In this manuscript Melnikov et. al., present data to show that Microsporidia carries same set of ribosomal proteins as other non-parasitic eukaryotes. They further show that, some of the ribosomal proteins no longer associate with ribosome but still they are preserved from genome decay due to their extra-ribosomal functions. They use Genome sequence analysis, non-parasitic ribosome structure analysis, and mass spectrometry to support their conclusions.

This is an interesting article and it shows how a minimal ribosomal unit is fully functional for Microsporidia. Their study shows that, despite reduced genomes in the parasites, some ribosomal proteins which have lost their roles in protein synthesis are still conserved and functional due to their non-ribosomal roles.

Thank you for these comments, especially for drawing our attention to the drawbacks in our illustrations. We have addressed the comments as following:  

This manuscript may be accepted in its present form. Few minor comments as follows;

Figure 2 should be removed from the Materials and methods section and should be put after Figure 1.

            The figure was moved to where it belongs in the Results section.

Figure labeling legends particularly in Figure 2 & 3 are way too small.

            The figures were reformatted to increase their size and label size.